# A standardized genome architecture for bacterial synthetic biology (SEGA)

Carolyn N. Bayer [1,2], Maja Rennig [1,2✉], Anja K. Ehrmann [1,2] & Morten H. H. Nørholm [1✉]

Chromosomal recombinant gene expression offers a number of advantages over plasmid-based synthetic biology. However, the methods applied for bacterial genome engineering are still challenging and far from being standardized. Here, in an attempt to realize the simplest recombinant genome technology imaginable and facilitate the transition from recombinant plasmids to genomes, we create a simplistic methodology and a comprehensive strain collection called the Standardized Genome Architecture (SEGA). In its simplest form, SEGA enables genome engineering by combining only two reagents: a DNA fragment that can be ordered from a commercial vendor and a stock solution of bacterial cells followed by incubation on agar plates. Recombinant genomes are identified by visual inspection using green-white colony screening akin to classical blue-white screening for recombinant plasmids. The modular nature of SEGA allows precise multi-level control of transcriptional, translational, and post-translational regulation. The SEGA architecture simultaneously supports increased standardization of genetic designs and a broad application range by utilizing well-characterized parts optimized for robust performance in the context of the bacterial genome. Ultimately, its adaption and expansion by the scientific community should improve predictability and comparability of experimental outcomes across different laboratories.

---

[1] Novo Nordisk Foundation Center for Biosustainability, Technical University of Denmark, Kgs. Lyngby, Denmark. [2] These authors contributed equally: Carolyn N. Bayer, Maja Rennig, Anja K. Ehrmann. ✉email: rennig@biosustain.dtu.dk; morno@biosustain.dtu.dk

Historically, recombinant plasmids have been crucial for handling DNA as they enable amplification and recombination of genetic parts for exploring biological systems. Vector collections such as the comprehensive pET plasmid series for recombinant protein production are widely used, and tricks for handling these, such as blue-white screening using beta-galactosidase as a genetic reporter, are today standard material in molecular biology textbooks. However, as the biological fields advance, strategies for chromosomal gene expression become increasingly important. Plasmids have inherent disadvantages, like constraints on the size of the hosted genetic elements and copy-number differences on the single cell population level[1], while integration in genomes provides higher intrinsic stability and lower metabolic burden to the host cell[2,3]. In addition, chromosomal integration circumvents the use of antibiotics, which reduces the costs in large-scale fermentations and the risk of spreading antibiotic resistance[4,5]. By reducing the burden to the host cell, reproducibility of experiments will increase and thereby the fundamental understanding of biology itself.

Following this trend, numerous technologies for integrating DNA constructs into the genomes of bacteria have been developed[6–13]. Typically these technologies use phage-derived homologous recombination systems, such as the λ-Red system from bacteriophage λ[7,9,11] or the RecET system from the Rac prophage, and antibiotic resistance markers are utilized to select for successful genomic integrations[14,15]. More recently, CRISPR/Cas9-mediated counter-selection was developed for scarless genome engineering in bacteria[10,12,13,16].

Driven by the vision to optimize biological systems and to incorporate engineering principles in their design, the field of synthetic biology emerged. With these new principles being applied to molecular biology, including abstraction of a complex system into parts with defined functions, the need for standardization became more apparent and the synthetic biology community accepted the lack of standards as a problem and started to advocate for their implementation[17–20]. While on one hand standardization could restrict flexibility and creativity, it will on the other hand enable more predictable performance of biological designs. Furthermore, standards enable automation and thereby can accelerate biological innovation.

Several attempts have been made to achieve standardization within the field[18] and organizations, such as Bioroboost (https://standardsinsynbio.eu), have been established with the main purpose to promote standardization in synthetic biology. With regards to plasmids, several standards have been described: BioBrick assembly marked the first approach of creating modular genetic parts for sharing between different laboratories[21]. This was followed by optimized versions[22–24] and included a growing repository of biological parts driven by the international genetically engineered machine (iGEM) student competition (http://parts.igem.org/). In a more recent approach, the Standard European Vector Architecture (SEVA) was created[25–27]. This vector collection is based on a simple three-component plasmid architecture, a standardized nomenclature, and a methodology that allows for easy interchange of backbone modules. Nevertheless, according to a recent study, most molecular biology standards are still in the innovator or early adopters' phase—and the field requires new impulses and simpler technologies that will convince an increasing number of researchers to adopt them[18].

While it is common practice for engineering of eukaryotic cells to integrate DNA constructs into the chromosome, methods are not standardized when it comes to genome engineering in bacteria. Plasmid-based approaches still dominate synthetic biology in bacteria, likely due to complicated genome engineering protocols and the need for multiple helper plasmids, strains, and DNA fragments—for example, three helper plasmids were necessary in one of the early protocols that combined λ-Red recombineering with CRISPR selection[10] whereas other methods required at least two rounds of genome engineering for integrating non-selectable fragments[15].

The work and resources described here, represent efforts to design a highly simplistic, standardized genome engineering platform that allows for predictable expression levels and rapid interchange of functional modules. Inspired by the SEVA plasmid series, the Standardized Genome Architecture (SEGA) enables simple and efficient swapping of functional modules on the genome. This is made possible by λ-Red phage-derived homologous recombination and by connecting the modules with selection and counter-selection markers using genetic landing pads integrated on the genome. Landing pads are a common concept in synthetic biology and have previously been used for example to aid in high-level gene integration in *Saccharomyces cerevisiae*[28] and as target sequences for mutant library generation on the genome of *Salmonella enterica* and *Escherichia coli*[29]. Other landing pads in *E. coli* contain recognition sites for phage integrases or endonucleases to aid an efficient recombination process[30–32]. Usually, these approaches still require a previous cloning step to assemble a promoter, a gene-of-interest, and suitable selection markers on a plasmid. The standardized design of SEGA facilitates the integration of, e.g., a gene-of-interest with high efficiency and predictable expression control across the genome. Apart from an active recombination system like λ-Red, no additional genetic elements are needed to achieve a final, antibiotic resistance marker-free construct in one simple step. This way, SEGA should expedite the shift from plasmid-based synthetic biology towards more predictably performing chromosomal designs.

## Results

**Philosophy and design of SEGA**. In the Standardized Genome Architecture (SEGA), genomic integration of DNA fragments is enabled by λ-Red recombineering and landing pads that contain features that (i) enable insertion of additional genetic elements and (ii) provide well-characterized functional parts such as promoters and genes, and (iii) provides insulation against genome context-dependent effects. The SEGA landing pads allow for reusable homology regions and time-efficient construction of parallel genetic designs with a minimal number of reagents and handling steps. SEGA strains can be stored as ready-to-use competent cryostocks with the recombineering machinery activated. This heavily simplifies the genome engineering process for the end-user. SEGA bricks are integrated on the genome simply by combining the two reagents (i.e., competent cells and DNA), followed by incubation steps, and successful recombinants are identified by visual inspection on agar plates. Thus, DNA can be prepared by a simple PCR amplified directly from original source or even simpler by ordering inexpensive synthetic DNA fragments. These steps do not rely on prior DNA assembly on plasmids, which may be problematic as some genes are toxic when handled on cloning vectors[33].

A typical engineered function in DNA includes a gene and genetic control elements that ensure the expression of this gene. We adopt the term cargo, defined as the DNA portion that bears the main functionality of the vector[25], but separate the control elements from the cargo: a SEGA landing pad typically hosts two major genetic control elements that influence gene expression on the transcriptional (C1), and translational (C2) level (Fig. 1). The SEGA strain collection presented here comprises different sets of experimentally validated promoters (with their regulators) as C1 and different translation initiation regions (TIRs) as C2 (Fig. 1). The TIR spans from the Shine-Dalgarno sequence to the 5th codon of the gene of interest[34].

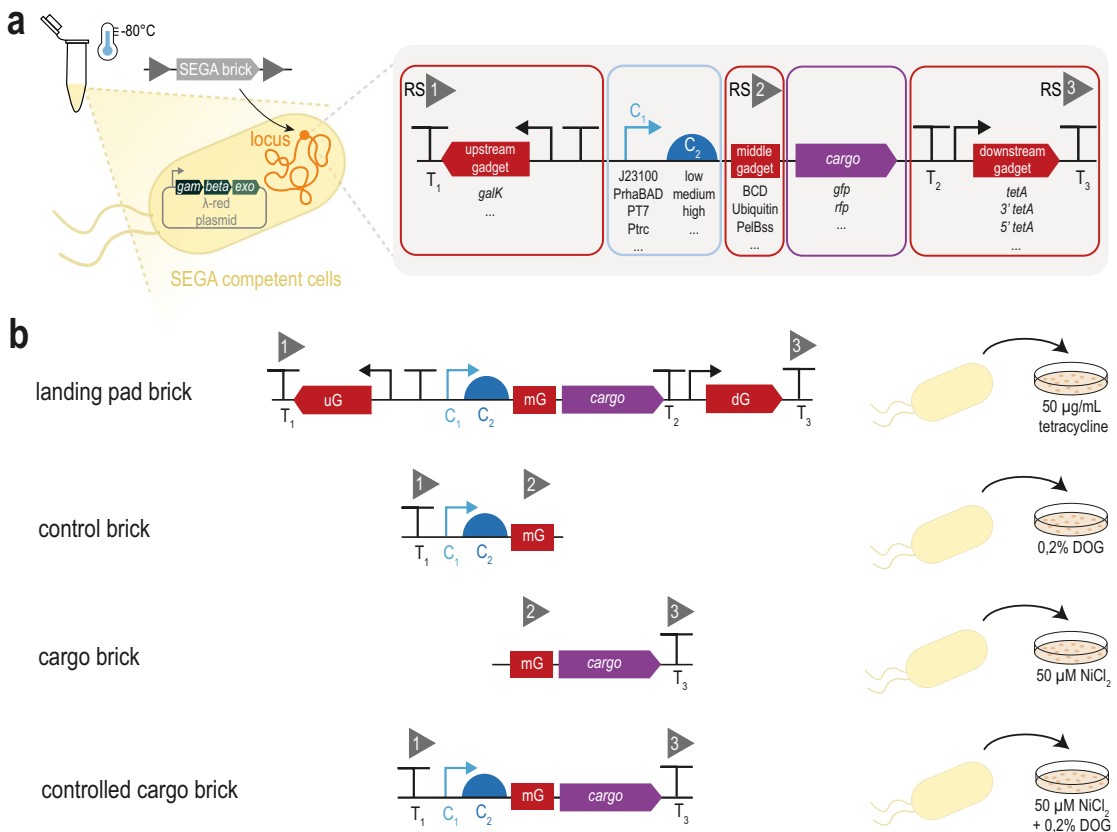

**Fig. 1 Architecture and engineering of SEGA. a** Illustration of the simplistic SEGA genome engineering approach. A DNA fragment—SEGA brick—is mixed with bacterial cells that contain SEGA landing pads and active recombination systems like λ-Red. The SEGA landing pad hosts control elements (blue color) for transcription (C1) and translation (C2), a cargo (purple) encoding the main genetic functionality, and additional gadgets (red color) harboring genetically insulating terminators, markers for selection and counter-selection, or elements for post-translational control (middle gadget). **b** The *galK* upstream gadget confers sensitivity to 2-deoxy-galactose (DOG) and facilitates integration of SEGA "control bricks". The *tetA* downstream gadget confers resistance to tetracycline and sensitivity to $NiCl_2$ and enables both landing pad brick and controlled cargo integration and cargo brick exchange. Gadgets also serve as standardized recombination sites (gray triangles, RS1–3). Depictions of the standard parts are compliant with the Synthetic Biology Open Language (SBOL) visual standard[74] (https://sbolstandard.org/visual/). mG middle gadget, uG upstream gadget, dG downstream gadget, DOG 2-deoxy-galactose, RS recombineering site.

We further introduced a number of gadgets (another concept adopted from SEVA) defined as dispensable DNA sequences that confer new utilities or properties to the basic frame of the vector[26]. For example, to minimize polar effects on host genes and vice versa, insulating terminators flanking the landing pad were included from a comprehensive, experimentally characterized collection[35]. Further, to aid the recombineering of parts, genetic markers for selection and counter-selection were added (Fig. 1a). A third gadget in between the control elements and the cargo provides additional features such as a bicistronic gene expression buffer[36], signal peptides, or peptide tags for, e.g., protein solubilization, detection, or purification. Since these gadgets contain a start codon and are closely interconnected with the C2 control elements, their sequences are key for translation initiation, and can standardize translation levels as outlined below.

Beyond the described functions, the gadgets facilitate modulization by constituting standard recombineering sites (RS1, RS2, RS3, Fig. 1; sequences can be found in Supplementary Table 4) that can be reused for recombination of different parts we term SEGA "bricks" (Fig. 1b).

**SEGA enables highly simplistic genome engineering with green-white screening.** An important function of the SEGA gadgets is to enable efficient and simplistic recombination of genetic parts, in addition to providing insulation from the genomic context with efficient terminators. The *tetA* gene is particularly suited for this task since it can be used for both positive selection, by conferring resistance to tetracycline, and counter-selection because the expression of *tetA* leads to sensitivity toward metal ions like $Ni^{2+}$[37]. Our standard SEGA landing pad contains, in addition to the control elements, a preliminary cargo encoding GFP, with the rationale that exchange of this cargo with any other DNA fragment would lead to the loss of GFP fluorescence enabling simple "green-white-screening" by visual inspection (Fig. 2a).

Fluorescence was exploited to test the performance of the downstream *tetA* gadget. A SEGA cargo brick, encoding the red-fluorescent protein *mCherry*, was integrated into the landing pad using homology regions RS2 and RS3 (Fig. 2b). Cryostocks of electrocompetent cells, with the λ-Red recombination machinery expressed[14], were thawed and mixed with or without the *mCherry* fragment. After recovery, the cells were plated on M9 agar supplemented with or without $NiCl_2$ (Fig. 2c). As expected, a green-fluorescent bacterial lawn could be observed in the absence of negative selection. When adding $NiCl_2$, far fewer green cells could be observed and with the addition of the *mCherry* fragment, mainly red fluorescent colonies were present. The efficiency for integrating *mCherry* varied between 80 and 100%

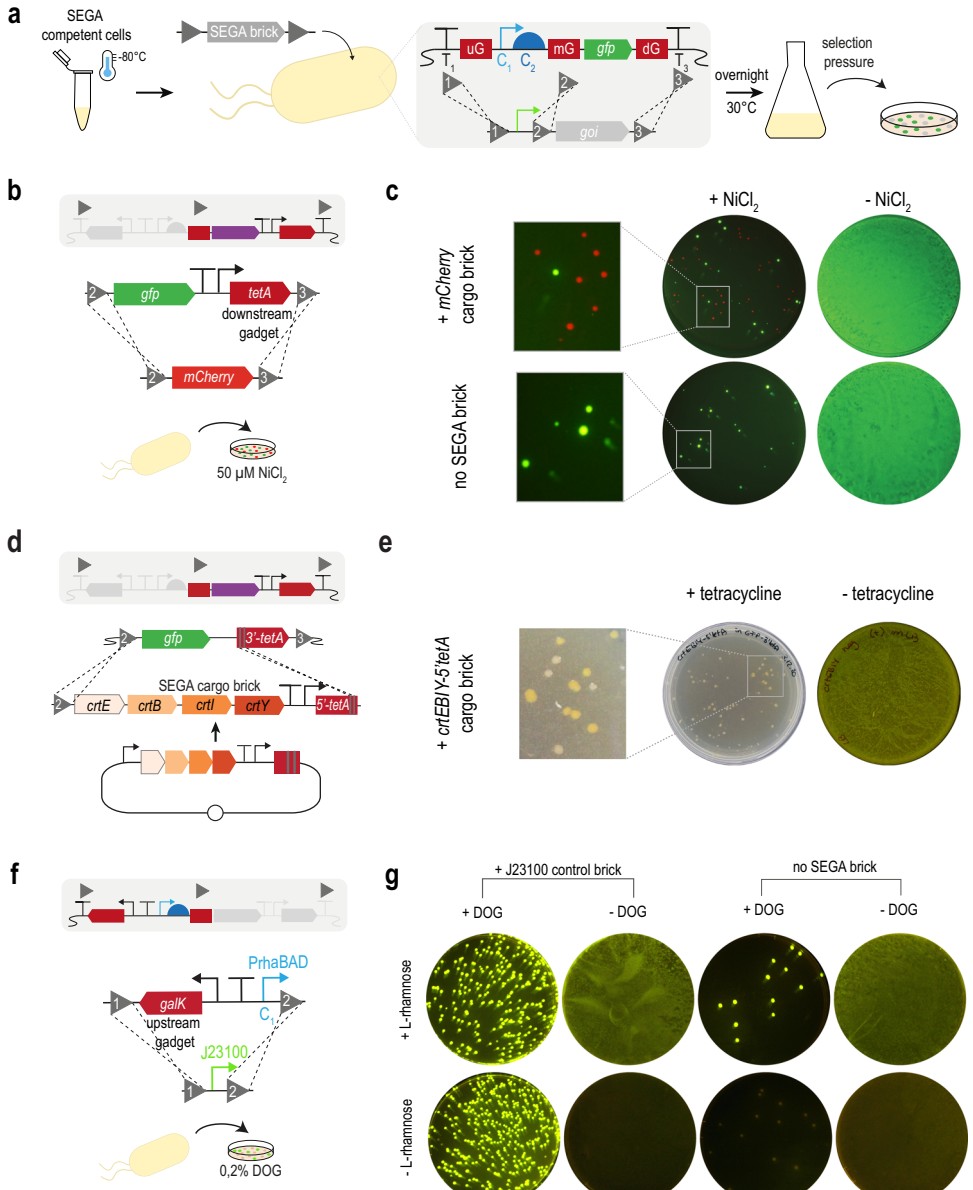

**Fig. 2 SEGA gadgets enable selection of recombinant bacteria with green-white screening. a** Illustration of SEGA green-white screening. Electrocompetent SEGA cells harboring active λ-Red proteins can be stored at −80 °C as a glycerol stock. The SEGA brick DNA fragment harboring the *gene of interest* (*goi*) is mixed with the thawed cells for electroporation. Depending on the transformed DNA, recombineering can occur at recombination sites 1–3 (grey triangles) to exchange either *gfp* or the control elements (C1, C2). The cells are then recovered overnight and plated on NiCl₂ or 2-deoxy-galactose (DOG) for counter-selecting against the gadgets. Recombinants are identified by loss or acquisition of green fluorescence. **b** A SEGA landing pad was constructed with a downstream *tetA* gadget and a constitutively expressed *gfp* cargo to enable exchange with *mCherry* by NiCl₂ counter-selection. *mCherry* was amplified by PCR with homology to recombination sites 2 and 3 (grey triangle). **c** the *mCherry* DNA was electroporated into a SEGA cells that were plated on M9 agar with and without NiCl₂. **d** A SEGA cargo brick containing the 4863 bp long full *crtEBIY* pathway and the 5'*tetA* were first assembled on a plasmid and then amplified for integration at RS2 and the split-site of the truncated *tetA*. **e** Integration of the full *crtEBIY* pathway into a SEGA landing pad using the truncated *tetA* gadget. Recombinants exhibit an orange color due to β-carotene expression **f** An upstream *galK* gadget was designed for simple exchange of control elements. **g** The PrhaBAD inducible promoter was exchanged for the constitutive promoter J23100 using an oligonucleotide with homology to RS1 and RS2. Cells were transformed with and without the ssDNA brick and subsequently plated on M63 agar supplemented with and without DOG or L-rhamnose.

and was more efficient with a gel-purified PCR fragment, likely due to the absence of interfering PCR primers (Supplementary Fig. 1). This shows that DNA fragments can be integrated into SEGA landing pads with high efficiency by simply adding DNA to frozen SEGA competent cells, creating a final, antibiotic marker-free construct on the genome in a single step. More commonly, successful recombinants will be colorless instead of red. Therefore, we termed this method "green-white-screening" designs, we encountered a technical challenge with *tetA*.

and validated the efficiency and easy identification by integration of another gene encoding a commercially interesting camelide-derived single domain antibody (Nanobody). We obtained between 81% and 94% white (positive) colonies across three replicates for the integration of the Nanobody gene (Supplementary Fig. 2).

While assembling SEGA landing pads and other complex designs, we encountered a technical challenge with *tetA*.

Occasionally, we assembled different DNA fragments on a plasmid followed by sequence verification, PCR amplification, and genome integration. However, *tetA* gadgets optimized for the genome appeared toxic to bacterial cells when handled on plasmids, probably due to the higher gene copy number. To address this problem, we devised an alternative approach that includes only the 5′ end of *tetA* including the promoter. This truncation can be handled on low-copy plasmids and *tetA* is restored on the genome when the complementing 3′-end is present there. We found this approach highly useful and efficient for introducing landing pad bricks, including different control elements and cargos assembled on plasmids (Supplementary Figs. 3a, 6a). We further tested the truncated *tetA* approach for the integration of the 4542 bp operon *crtEBIY* from *Pantoea ananatis* encoding four genes necessary to produce the orange food coloring pigment β-carotene (Fig. 2d). Previously, our attempts to integrate this cargo brick with counter-selection against *tetA* were unsuccessful, probably due to the large size of the DNA fragment. We assembled the *crtEBIY* pathway together with the 5′*tetA* fragment and its accompanying control elements on a plasmid, amplified the whole construct by PCR, and subsequently integrated it in one step. The green-white screening showed no false positive (green fluorescent) clones while the orange color helped estimating that over 70% of the obtained colonies carried a functional *crtEBIY* operon (Fig. 2e, Supplementary Fig. 3d). Analysis of the remaining colorless clones showed sequence errors that presumably rendered the pathway non-functional.

The truncated *tetA* gadget increases the integration efficiency of large SEGA bricks, but also provides the basis for consecutively building larger and more complex synthetic biology designs on the genome, by cycling between *tetA* selection and counter-selection. As a simple demonstration of how the dual-selection marker can be used for sequential genome engineering, we chose to split the *crtEBIY* pathway into the four individual genes and integrated them in a stepwise manner. We cycled through four rounds of selection and counter-selection to introduce the complete operon (Supplementary Fig. 3b, c). In the first and third round of recombineering, *tetA* was truncated, whereas in the second and fourth round, functional *tetA* was reconstituted. This demonstrates successful recycling of a selection marker with minimal technical effort for building advanced synthetic biology designs in multiple rounds of genome engineering.

In *E. coli*, *galK* can be used as a positive selection marker on galactose as well as a counter-selectable marker by adding the toxic galactose analog 2-deoxy-galactose (DOG)[38]. In the SEGA landing pad, a *galK* upstream gadget enables the exchange of control elements such as promoters. For this purpose, the native *galK* gene was deleted from the genomes of selected strains of the SEGA collection. To demonstrate the functionality of the *galK* gadget, we constructed a landing pad that contained *galK* and the rhamnose-inducible PrhaBAD promoter driving expression of *gfp*. This way, when the C1 element is exchanged for a constitutive promoter, recombinant genomes are identified as green in the absence of L-rhamnose (Fig. 2f), whereas recombinant non-constitutive promoters are identified as white in the presence of rhamnose (Supplementary Fig. 4). SEGA competent cells were electroporated with oligonucleotides encoding the constitutive J23100 promoter and plated on M63-glycerol agar with or without DOG and L-rhamnose. In the presence of DOG and the J23100 oligonucleotide and the absence of L-rhamnose, 95% of the colonies were fluorescent, indicating successful exchange of C1 (Fig. 2g). By omitting the selection pressure of DOG, a bacterial lawn could be observed that was fluorescent only when L-rhamnose was present. This shows that the *galK* gadget can be used to successfully exchange SEGA control elements with high efficiency by simple addition of oligonucleotides or PCR products and that different recombinant promoters can be identified by green-white screening.

## Construction and validation of a SEGA strain collection with different control elements.

We envision SEGA to be an extraordinary simple, but also comprehensive and versatile platform. To this end, we engineered a range of the most commonly used control elements into SEGA landing pads allowing streamlined construction of multiple variants encompassing all combinations of C1 and C2 by adding just a single cargo brick (see Figs. 3 and 4). The current SEGA collection of more than 100 strains is listed in Supplementary Table 1 and Supplementary Data 1. We included four promoters that have been characterized extensively before: the strong constitutive promoter J23100 from the Anderson collection and three inducible promoters: PT7, PrhaBAD, and Ptrc[39–41]. These were all placed on the genome together with a GFP cargo and the *tetA* gadget. The performance of the four promoters in the landing pad context was tested by following growth and fluorescence in a microplate reader (Fig. 3a). Ptrc is highly inducible by IPTG, but also provides a basal level of ("leaky") transcription in the absence of the inducer. PrhaBAD is completely off in the absence of the inducer, and the level of expression is titratable by the addition of L-rhamnose. Expression of *gfp* from PT7 is the strongest among the tested C1 elements, but it is tightly repressed in the absence of IPTG. For all tested promoters, the observed performance was in good agreement with their established performance in plasmid systems.

We additionally built landing pads with two inducible promoters from the "Marionette" collection as C1 elements: PSal and PTet[42]. This allowed a simple test of the effect of regulator position on the performance of inducible promoters in SEGA: either the regulator was arranged divergently to the promoter within the landing pad or it was present in a distant genome location in a regulator array as described by Meyer et al. The response function of PSal differed mostly in terms of maximum expression strength but not in sensitivity of the regulator between the two designs (Supplementary Fig. 5). Furthermore, the performance of Ptet in SEGA is severely affected by the presence of the tetracycline efflux pump expressed from the *tetA* gadget, but expression of *gfp* can be induced as long as the *tetR* regulator is integrated as part of the landing pad.

Besides transcription, translation is a major determinant of gene expression, and the ability to manage both provides highly stringent control of synthetic biology designs[43]. To address this, we engineered C2 control elements—translation initiation regions (TIRs)—known to control the speed of translation initiation with a major impact on overall translational efficiency[44,45]. In *E. coli*, the TIR covers both the Shine-Dalgarno sequence and the 5′end of the expressed gene[46–49]. Thus, the TIR would typically be part of the incoming cargo, which precludes standardization of translation strength across different cargos. For this reason, we included a third gadget position between the control elements and the cargo that can serve multiple purposes: it provides a recombination site independent of C1 and C2, it encodes a standardized TIR independent from the cargo, and it can provide post-translational control and facilitate downstream applications. We picked a range of gadgets that offer useful features in the 5′end of various cargos: a bicistronic design (BCD) that will leave proteins of interest untagged, while still permitting normalization of expression levels[36]; a ubiquitin-his tag for solubilizing proteins that at the same time enables histidine-tag mediated purification[50], a TEV protease site providing the opportunity to release a custom protein N-terminus[51], a PROTi tag affecting the stability of proteins on demand[52]; and two different secretion signals, encoding for a PelB signal peptide (PelBss) and YebF protein, serving the role to target proteins of interest to the periplasm or to secrete them outside of the cell[53].

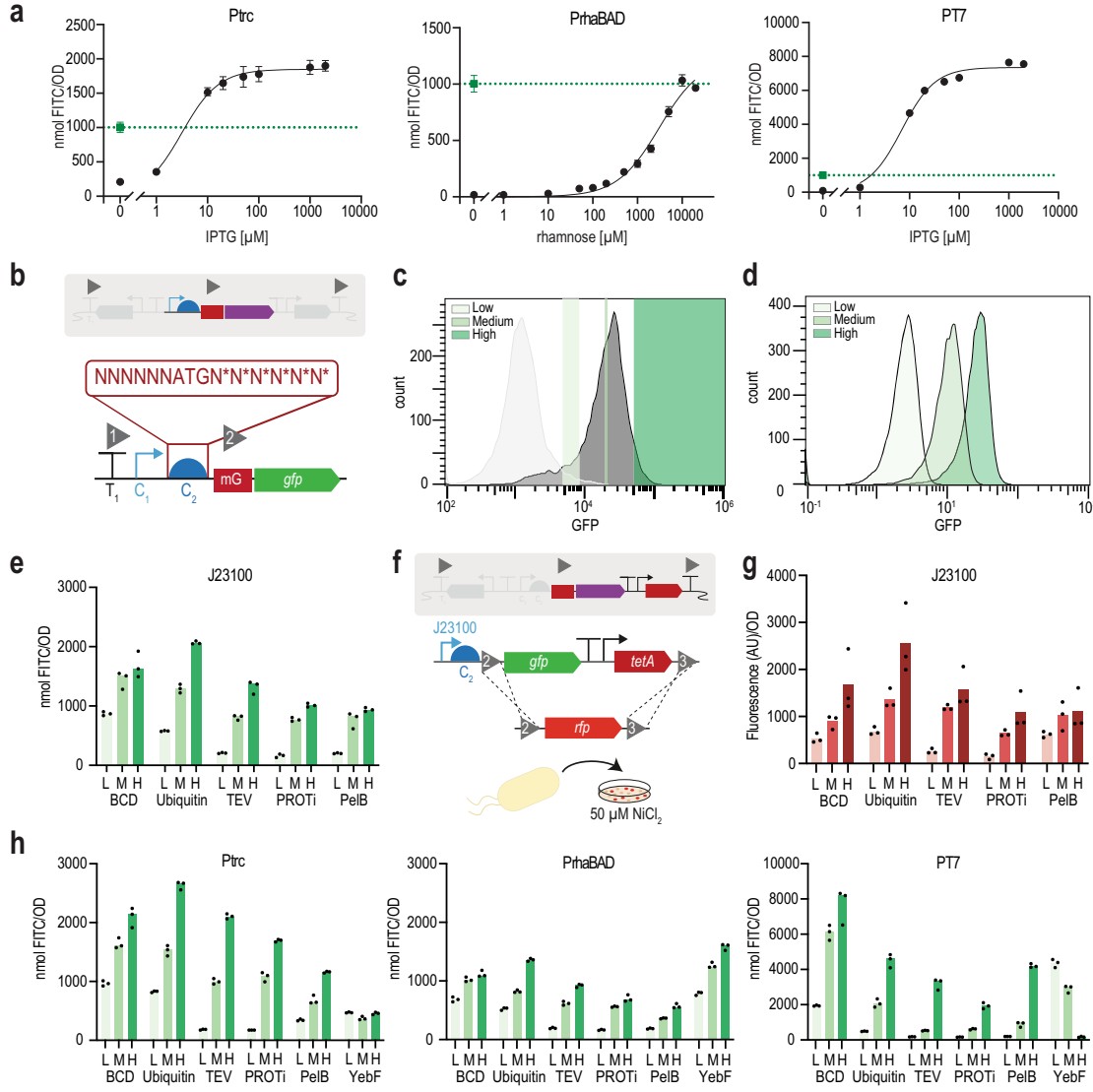

**Fig. 3 Assembly and characterization of different SEGA control elements. a** Performance of the different C1 elements included in SEGA was tested in a fluorescence microplate reader with different concentrations of IPTG (Ptrc and PT7), and rhamnose (PrhaBAD). The expression strength of the constitutive promoter J23100 is shown as an internal reference (green line) in the three plots. Data points represent the means of three biological replicates with standard deviations. **b** To obtain C2 control elements with different translational efficiency (low, medium, and high), translation initiation region (TIR) sequence libraries were generated for a range of different gadgets that harbor specific functions (see main text and Supplementary Fig. 6 for further details). **c** Low, medium, and highly expressing variants were chosen for each gadget with J23100 as C1 and sorted via FACS. The library of PelBss is shown as a representative example (see also Supplementary Fig. 6). **d** Expression levels from individual clones were confirmed by flow cytometry (representative example here C1: J23100, PelBss gadget, see also Supplementary Fig. 7) and by **e** measurement in a microplate reader. Data shown in panel **c** and **d** were obtained from a single experiment. **f** To show the transferability of the TIRs to other cargos, *gfp* and *tetA* were replaced by *rfp* using homologous recombination at recombination site 2 and 3 by selection on NiCl₂. **g** Fluorescence values of these strains were obtained in liquid LB medium. **h** The selected TIRs were then transferred to the other C1 elements (PrhaBAD, Ptrc, and PT7) and fluorescence was assessed. In panels **e**–**h**, the bars represent the mean of three biological replicates. The datapoints of the three individual replicates are shown. In all panels, GFP fluorescence was normalized to a fluorescein standard curve and OD₆₃₀ values (see Supplementary Fig. 11). Source data are provided in the Source data file.

To provide experimentally validated TIRs with defined translation levels for these different gadgets, we first constructed TIR sequence libraries, randomizing six nucleotides upstream from the start-codon as well as changing the first two codons after the ATG to all synonymous codons (Fig. 3b). This is an established and simple approach that can create variation in expression levels by more than 1000-fold[45]. The TIR libraries were introduced into a landing pad with the constitutive J23100 promoter and the truncated 3'*tetA* gadget. The gadget variants were assembled on plasmids together with *gfp* and the 5'*tetA* fragment. The constructs were then amplified by PCR, using degenerate oligonucleotides to randomize the C2 elements, and integrated to complete the truncated *tetA* sequence on the genome (Supplementary Fig. 6a). The resulting TIR libraries were grown in liquid culture to mid-exponential phase and separated into three *gfp* expression levels—low, medium, and high—by fluorescence-activated cell sorting (FACS) (Fig. 3c, Supplementary Fig. 6b–g). Single clones were grown and selected from agar plates, purified by restreaking, and validated by sequencing (the TIR sequences can be found in Supplementary Table 5). The expression levels were subsequently verified by growth and fluorescence in a microplate reader and flow cytometer (Fig. 3d, Supplementary Fig. 7).

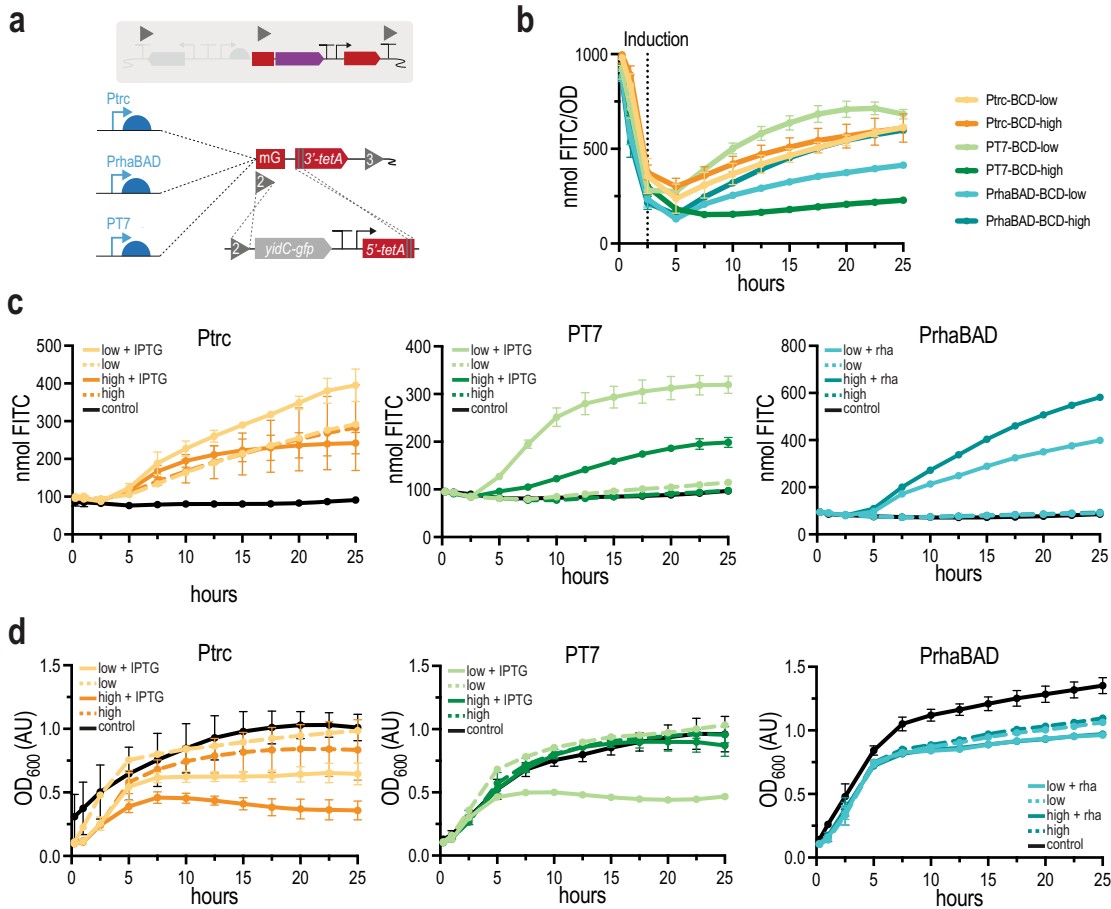

**Fig. 4 Production of the toxic protein YidC-GFP with different SEGA control elements. a** To show the benefit of simultaneous integration of one PCR product in several SEGA strains, *yidC-gfp* was integrated into six different SEGA strains with different C1 control elements with low and high TIRs and a BCD gadget using homologous recombination at recombination site 2 and the truncated *tetA* gadget. **b** Production of YidC-GFP (nmol FITC) in six different SEGA strains was measured over 25 h and normalized to cell density. Cells were induced after 2.5 h. **c** Absolute production of YidC-GFP (nmol FITC) was measured for 25 h for all constructs. **d** Growth (OD$_{600}$) was measured for 25 h for all constructs. All measurements represent the mean of three biological replicates with standard deviations, except the sample Ptrc-low uninduced (data obtained from a single experiment). Source data are provided in the Source data file.

In our SEGA designs, the C1 and C2 sequences are always separated by an identical 23 bp sequence allowing the recombination of all different promoters and gadgets with different TIRs. This enabled the amplification of different C2 and gadgets together with *gfp* and 5′*tetA* from the FACS-sorted strains and integration into strains harboring the different C1 elements and the 3′*tetA* gadget (Supplementary Figs. 8a–10a). This way, we constructed all combinations of C1, C2, and gadget elements and verified the GFP expression levels for the resulting strains in a microplate reader and flow cytometer. Overall, we observed that the different C2-gadgets preserved the expression strength predictably at low, medium, and high levels when recombined with a different promoter (Fig. 3e, Supplementary Figs. 8–10). However, the total expression levels varied as expected depending on the promoter. One exception was the YebF gadget: the expected expression pattern was only observed in combination with PrhaBAD and we were unable to construct SEGA strains combining the J23100 promoter with YebF for extracellular secretion. The YebF strains were functional with Ptrc and PT7, but the expression strength deviated from the predicted low, medium, and high levels (Fig. 3h).

The main application of SEGA will probably be the simple exchange of cargos. Thus, it is relevant to test if different control elements and gadgets offer predicable expression of different cargos.

To test this, *gfp* and *tetA* were replaced by *rfp* in all J23100 landing pads with different C2 gadgets (Fig. 3g) and performance of all constructs followed the same trend in expression levels as previously observed for *gfp* (Fig. 3e). This demonstrates an overall predicable performance of the SEGA modules with different cargos.

**Multi-level control of a difficult-to-express gene is enabled by SEGA.** The interplay of transcriptional and translational control is critical when it comes to gene expression burden or product toxicity[43]. The standardized setup and the simplistic genome engineering workflow of SEGA with multi-level control might allow for optimizing genomic constructions with targets that show toxic effects. To explore this, we integrated the gene encoding the *E. coli* membrane protein YidC fused to GFP, which is known to be toxic when overexpressed[54]. With the SEGA collection, only a single DNA fragment encoding *yidC-gfp* flanked by sequences homologous to RS2 and the truncated *tetA* gadget was required to construct six different production strains (Fig. 4a). The Ptrc, PT7, and PrhaBAD promoters were combined with the C2: low or C2: high translational control elements. We observed that integration into the Ptrc background yielded much fewer colonies, but positive recombinants were found for all

combinations. Single clones were restreaked and validated by sequencing, but all sequenced PT7 constructs with a high TIR had a mutation in the T7 promoter region. Growth and expression were measured in a microplate reader. Based on fluorescence normalized to cell density, more YidC-GFP was produced under control of PT7 with the low translational control element than with the high TIR (Fig. 4b). We observed leaky production from Ptrc resulting in quite high protein titers without induction, but again the low TIR was performing better than the high TIR, when production was induced with IPTG (Fig. 4c). Moreover, induction of YidC-GFP expression from Ptrc and PT7 with IPTG resulted in arrested growth (Fig. 4d). However, the constructs under control of PrhaBAD produced the highest absolute amount of YidC-GFP (Fig. 4c) and there was no clear correlation between production levels and cell growth (Fig. 4d). This demonstrates that SEGA strains with different control elements can easily be screened simultaneously to identify the optimal production scenario for a challenging target.

**Efficiency of integration of SEGA landing pads varies between different genomic locations**. One of the first decisions to make when introducing new features on a genome is the target locus and this may affect both cellular fitness and the performance of the construct. To study SEGA landing pad integration in different locations, we chose 11 previously characterized loci evenly distributed across the *E. coli* K12 MG1655 genome[55–57] (Fig. 5b). The landing pad was typically placed directly downstream of a native gene, when possible in an intergenic region between two converging genes, to avoid disruption of promoters and other regulatory sequences (Supplementary Table 6).

We observed that some genomic locations seemed more difficult to target for manipulations than others. Similar observations were previously reported in a study targeting the whole genome with a transposon-based approach[57]. Across 11 different loci, integration efficiencies ranged from 2 to 500 colony forming units (CFU) per μg DNA with the majority in the low-efficiency range (Fig. 5c). Possibly, different homology sequences used for targeting different loci could play a role in determining the integration efficiency. To control for this, we integrated the landing pads in different K12 sibling strains from the Keio collection[58] harboring identical *kanR* cassettes in different genomic locations equivalent to the loci targeted before. This enabled the use of identical homology sequences, i.e., the exact same DNA construct could be integrated in different loci (Fig. 5a). However, we still observed a wide distribution of integration efficiencies for different loci (Fig. 5c), and there was poor correlation between the integration efficiencies for the intergenic integration and the integration into the equivalent Keio strain. (Supplementary Fig. 12).

We then tested the integration efficiency of a SEGA cargo brick, in this case *rfp*, into the different genome locations harboring the SEGA landing pad (Fig. 5d). Selection for the correct integration was performed via NiCl$_2$ counter-selection and similar to what was observed previously, the integration efficiencies were distributed over a wide range, but the number of colonies obtained with the SEGA integration approach was sufficient to screen multiple clones in all tested loci (Fig. 5d). Due to the design of the SEGA landing pad, it is easy to differentiate false positive background colonies on the counter-selection plates by green-white-screening. We estimated the true positive rate of integration in 11 different genome locations by calculating the percentage of green and red fluorescent colonies across three replicates (Fig. 5e). The integration efficiencies into the SEGA landing pads, based on *tetA* counter-selection, were similar in most of the tested loci with true positive rates of above 80%. The

only exceptions were the *ycbX* and *wbbL* loci where a high background of green colonies was observed. This provides important experience with engineering into different genomic loci and validates the functionality of SEGA landing pads across the *E. coli* genome.

**Performance of SEGA modules in different genome locations**. It is already well-established that the genome location affects the expression strength of native genes as well as heterologous constructs[56,57,59,60]. We wanted to further investigate the effect of genome positioning on the performance of our SEGA landing pads and different modules such as C2 control elements. Non-disruptive integrations of the landing pads in 11 different genome locations were constructed as described above. For all integrations, a J23100 promoter landing pad brick was used, and for each locus two variants with low and high translation strength (C2) were constructed. The GFP expression of the 22 resulting strains was measured in late exponential phase in a microplate reader and the expression patterns across different loci followed the generally established notion that gene expression decreases with increasing distance from the origin of replication (Fig. 5f and g)[55–57,60]. However, we observed outliers from this general trend: an example is the *ykgH* locus where *gfp* expression was 4-fold diminished compared to expression from the *lacZ* locus, which is located only 41 kb away. Importantly, the low and high expressing variants of the control element C2 showed the expected low- and high-range expression levels across the genome. The median expression from the high translation control element was 3.1-fold higher that expression from the low translation control element (Fig. 5h), emphasizing the standardized modular performance of SEGA. Comparing the expression from the landing pad in different loci yielded a median fold change of 1.3 and 2.1, respectively, for constructs with either the C2: low or C2: high translation initiation control element (Fig. 5h). Again, this demonstrates the standardized performance of different SEGA modules and provides clues to the relative importance of different factors such as promoters, TIRs, genomic locations etc. on the performance of synthetic biology designs.

## Discussion

SEGA addresses two aspects that typically need to be considered when engineering biological systems: first, the compositional layout and abstraction of the system into parts, and second, the flow of information through all stages of the central dogma of molecular biology: from transcription to translation to the protein itself by incorporating gadgets that can affect protein stability, localization, solubilization, or downstream applications. Here, we have demonstrated that this multi-level control enables an overall predictable performance of genetic constructs and a screening tool for optimizing challenging genetic functions.

SEGA reduces genome engineering to an extremely easy task: SEGA cryostocks are simply mixed with DNA fragments ordered from a commercial vendor or generated in-house by PCR. For exchange of small DNA fragments such as promoters, this can even be accomplished with inexpensive single-stranded oligonucleotides. More complex designs can be assembled and sequence-validated on plasmids prior to genome integration based on the truncated *tetA* approach. Furthermore, the user can easily identify recombinants by employing green-white screening by eye.

The typical SEGA landing pad has a size-range of 2−5 kb and the efficiency of integration into different loci in *E. coli* varies greatly. We were able to rule out that this variation stems solely from the different homology sequences by integrating identical DNA fragments into 10 strains from the Keio collection[58], and by integrating *rfp* into strains, which already carried the SEGA

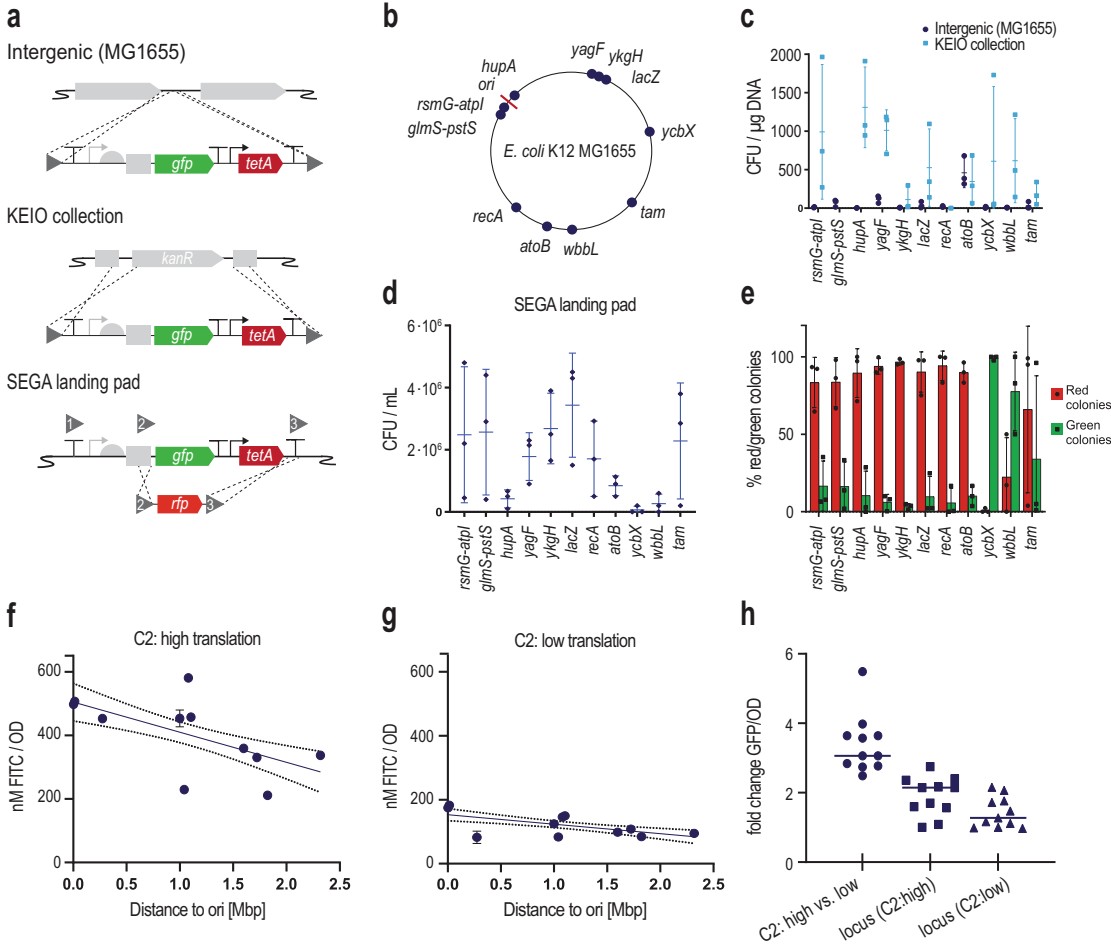

**Fig. 5 Chromosome position affects integration efficiency and gene expression. a** Engineering of 11 different chromosome locations was tested using three different integration methods. Non-disruptive, intergenic integration in K12 MG1655 was compared with exchanging the kanR cassette in the BW25113 derivates of the Keio collection and cargo exchange within the SEGA landing pad. All results shown in panels **f**–**h** are based on intergenic, non-disruptive integrations. **b** Overview of the 11 engineered genome locations. **c** Integration efficiency of intergenic integration as well as into the Keio collection, in which case the homology sequences are identical for all genomic locations. **d** Integration efficiencies of *rfp* into the SEGA landing pad across different chromosomal locations. Efficiencies are given in CFU per mL of the overnight recovery culture. **e** Ratio of true positive and false-positive colonies after recombination using the SEGA approach in different loci. **f** Constitutive GFP expression (C1: J23100, C2: high, BCD gadget) in 11 genome loci in late exponential growth phase in relation to the distance to the origin of replication (ori). The result of a linear regression analysis is shown with a 95% confidence interval. **g** GFP expression in 11 genome locations as in (**e**) but with C2: low. **h** The fold change of GFP fluorescence between C2: high and C2: low strains measured for 11 genome locations. The fold change in GFP fluorescence was also measured for each genome location compared to the lowest expressing locus with the same C2 element. The line represents the median fold change across 11 genome locations. In panels (**c**)–(**g**), the data represent the mean of three biological replicates with standard deviations. Source data are provided in the source data file.

landing pad in different genome locations. This demonstrates the advantage of using pre-characterized and ideally pre-engineered integration sites for genome engineering to avoid time-consuming iterations of low-efficiency recombination attempts. These efforts also confirmed the established perception that gene expression strength decreases with increasing distance to the origin of replication. The major deviation from this trend was the large difference in expression from the *lacZ* and the *ykgH* loci, which are located close together. This difference may be a result of transcriptional silencing of the *ykgH* locus through protein occupancy[57,61]. The 11 integration sites explored here do not necessarily represent optimized expression loci: recent studies have attempted to identify and characterize genomic integration sites with minimal disruption of the host expression that might be better suited for further optimization[32,60,62] and the relative expression strength of a genome location can depend on the growth media and carbon source[60]. Another recent study comprehensively characterizes multiple areas of transcriptional

silencing and enhancement[57]. These studies highlight the need for further investigations on the role of genome structure in gene expression.

Approaching synthetic biology with a standardization mindset provides important clues to the general performance and robustness of different genetic modules. For example, analysis of the BCD gadget revealed strong overlap of the expression profiles for the three FACS-sorted C2 TIR elements (Supplementary Figs. 7a, 8b–10b). This BCD element was selected from the library of bicistronic designs characterized by Mutalik et al. and includes a strong Shine-Dalgarno, within the BCD, driving the translation of the downstream cargo/gene[36]. One design goal of a BCD is to normalize translational strength regardless of the genetic context. This means that adjusting the translational level by upstream TIR randomization may be less effective because it is counteracted by the downstream TIR. Since the inclusion of a BCD offers the attractive possibility for standardizing the expression level for an un-tagged cargo, it might be useful to include other, low

expressing bicistronic designs from the BCD library to offer standardized low-expressing variants in SEGA.

The strains harboring *yebF* gadgets form another exception to the otherwise standardized and predictable performance of the SEGA control elements. We hypothesize that these effects can be linked to toxic effects of the expression of the full-size YebF protein (see Supplementary Fig. 11), which comes into effect both during constitutive or inducible expression from a strong promoter. Only the PrhaBAD promoter led to a predictable low, medium, and high expression with different TIRs and the *yebF* gadget. A better performing signal peptide for secretion outside *E. coli* would be a valuable future addition to SEGA. Similar observations were made with the toxic YidC: the PrhaBAD promoter performed robustly and produced YidC at predictable levels with no growth impairment. In contrast, the T7 promoter strains showed growth impairment and unpredictable TIR performance. It is possible that toxicity is linked to RNA rather than protein stress, as the apparent protein levels of the PrhaBAD BCD-high construct, based on GFP fluorescence, still exceeded those of the PT7 strains. This is supported by recent work showing that RNase E mutants compensate for YidC expression toxicity[63]. In any case, we find that the multi-level control enabled by the SEGA standard can circumvent toxic effects regardless of the underlying mechanism. Furthermore, SEGA can help identifying both individual genetic parts and module combinations that unexpectedly fail in a genetic context. By detecting this in an early stage of the engineering process, the number of iterations of the bioengineering design–build–test cycle can be reduced.

With SEGA we have tried to realize our vision to make genome engineering more accessible. We hope that the highly simplistic protocol by itself will facilitate its use. The strain collection is available for non-commercial use through the Belgian Co-ordinated Collections of Microorganisms (https://bccm.belspo.be), all sequences can be downloaded in Genbank format, and detailed protocols are available at protocols.io[64]. More information can also be found on www.sega-genomes.com.

## Methods

**Strains, cultivation, and media composition**. The strains used in this study are listed in Supplementary Table 1. *E. coli* NEB5α (New England Biolabs) was used for cloning of plasmids. *E. coli* SIJ19[65] was used for assembly of the initial landing pad. *E. coli* MG1655 (lab stock) and *E. coli* MG1655 (DE3)[66] were used for assembly of SEGA landing pads. *E. coli* MG1655 (lab stock) and *E. coli* BW25113 KO strains (Keio collection[58]) served as chassis for comparison of different genomic loci. The Marionette Sensor Collection was a gift from Christopher Voigt (Addgene Kit #1000000137[42]).

All bacterial strains were cultivated in lysogeny broth (LB) with shaking (250 rpm) at 37 °C, if not stated otherwise. Bacterial strains harboring pSIM19[67] were always cultivated at 30 °C. Media was supplemented with antibiotics when required. Unless otherwise stated, antibiotics were used in the following concentrations: spectinomycin (50 μg/mL), kanamycin (50 μg/mL), and tetracycline (25 or 50 μg/mL). Chemically competent cells of NEB5α were obtained with a standard $CaCl_2$ protocol. All other strains were made electro-competent using a standard protocol. For counter-selection with $NiCl_2$, M9 agar plates were used. M9 agar contained 2 g/L glucose, 1x M9 salts, 2 mM $MgSO_4$, 100 μM $CaCl_2$, 1X trace elements, and 0.5 μg/mL thiamine. 10X M9 salts consist of 68 g/L $Na_2PO_4$, 30 g/L $KH_2PO_4$, 5 g/L NaCl, and 10 g/L $NH_4Cl$. 1000X trace elements consist of 10 g/L $FeCl_3$ x6$H_2O$, 2 g/L $ZnSO_4$ x7$H_2O$, 0.4 g/L $CuCl_2$ x2$H_2O$, 1 g/L $MnSO_4$ x$H_2O$, 0.6 g/L $CoCl_2$ x6$H_2O$, 3.2 mL/L 0.5 M $Na_2EDTA$, pH8. For counter-selection with 2-Deoxy-D-galactose (DOG), M63 agar plates were used. M63 agar contained 0.2% galactose for positive selection or 0.2% glycerol and 0.2% DOG for counter-selection, 1X M63 salts and 1 mM $MgSO_4$. 5X M63 salts consist of 10 g/L $(NH_4)_2SO_4$, 68 g/L $KH_2PO_4$, and 2.5 mg/L $FeSO_4.7H_2O$.

**Plasmid construction**. All plasmid manipulations were performed using uracil excision cloning[68]. Plasmids used in this study are listed in Supplementary Table 2. Oligonucleotides were received from Integrated DNA Technologies (IDT, Coralville, IA, USA) (Supplementary Table 3). PCR was carried out with Phusion U Hot Start Polymerase (Thermo Fisher Scientific, Waltham, MA, USA) or in-house NeqXX7 DNA polymerase (not published). PCR products were visualized on 1% agarose gels using the iBright Imaging System (Thermo Fisher Scientific, Waltham,

MA, USA) running the iBright gel imager software (v1.6.0). PCR products were purified using the NucleoSpin® Gel and PCR Clean-up Kit (Macherey-Nagel, Düren, Germany). All plasmids were isolated using the NucleoSpin Plasmid Purification Kit (Macherey-Nagel, Düren, Germany).

For assembly of SEGA landing pads, pSEVA27-sl3 was used as backbone[69]. A construct consisting of the BBa_J23100 promoter, the BCD gadget, *gfp*, and the first 150 nucleotides of *tetA*, separated by a terminator and the P3 promotor (driving expression of *tetA*) was amplified from the genome of *E. coli* MG1655 TetA$^{OPT}$ (not published) and assembled on pSEVA27-sl3 using oligonucleotides #201–#204. Subsequently, plasmids featuring the other middle gadgets (TEV, PROTi, ubiquitin, PelBss, and YebFss) were constructed using oligonucleotides #205–#217. The ubiquitin gadget was amplified from plasmid pET39-Ub(His10)[50] in a two-fragment cloning using oligonucleotides #208–#211. Together with PelBss or YebFss, *sfgfp* was introduced in a two-fragment cloning step with oligonucleotides #212–#217. *sfgfp* was amplified from pET28a-*sfgfp* (provided by Cristina Hernández-Rollán). After selection of the YebF C2 TIR sequences using YebFss, the full-length YebF gadget was reconstructed on plasmids under control of PrhaBAD using oligonucleotides #220–#227. Simultaneously, a sequence stretch between *lacI* and the PT7 and Ptrc promoter was removed from pET28a-AraH-GFP[70] and pET-Duet1-Ptrc-GFPOpt (Ptrc)[68], respectively, with oligos #218–#219, in order to avoid sequence homology with *tetA*.

For construction of pSEVA27-sl3-crtEBIY-5′TetA, the pSEVA27-J23100-BCD-GFP-5′TetA backbone was amplified with #228 and #229. The *crtEBIY* operon was amplified from plasmid pSEVA36-sl3-crtEBIY[69]. For sequential integration, *crtB* and *crtIY* were cloned independently into pSEVA27-BCD-GFP-5′TetA using oligos #231–#234, resulting in pSEVA27-crtB and pSEVA27-crtIY, respectively.

For construction of pSEVA27-sl3-YidC-GFP-5′TetA, pT7-*yidC-gfp* was amplified from pET28a-YidC-GFP[71] using oligos #239 and #240. The pSEVA27-sl3-5′TetA backbone fragments were amplified using oligos #235–#238.

For the construction of PSal and Ptet landing pads based on the Marionette collection, the regulators from plasmids pAJM.771 and pAJM.011[42] were recloned using oligos #241–#246 to match the promoter architecture of the SEGA landing pad, yielding plasmids pAJM.771-NahR-PSal and pAJM.011-TetR-Ptet.

**Construction of SEGA landing pads**. For construction of the initial SEGA landing pad *E. coli* strain SIJ19[65] was used. A biscistronic design[36] was inserted upstream of *gfp* using λ-Red recombineering and CRISPR/Cas9 counter-selection. In a parallel study, we then integrated a *tetA* gene downstream of *gfp* and optimized the translation initiation region of a *tetA* gene under control of the P3 promoter resulting in *E. coli* MG1655 TetA$^{OPT}$. Simultaneously, to facilitate integration of the plasmid-derived landing pads into the genome, MG1655 TetA$^{OPT}$ was truncated using MAGE oligo #106 and $NiCl_2$ counter-selection. TetA was rendered non-functional by removing the promoter and the first 78 base pairs. In the same process, *gfp* was removed. Upstream of the J23100 promoter the L3S2P11 terminator[35] was integrated by homologous recombination. This resulted in MG1655 J23100-3′*tetA* (SEGA001).

Subsequently, variants of the SEGA landing pad were constructed on pSEVA27-sl3 as described above. Then the construct gadget-*gfp*−5′*tetA* was amplified with oligos #118–#123 and #111 for homologous recombination and integrated into SEGA001. Simultaneously, we randomized a part of the translation initiation region. Details are described in the section "Library design and construction". Since the PROTi gadget shares the 5′ sequence with the TEV middle gadget, the TIR sequences for the PROTi gadget were not selected from the PROTi TIR library, but instead the TIR sequences selected for the TEV gadget were incorporated into the J23100-PROTi-*gfp-tetA* strains with oligonucleotides #115–#117.

In order to construct strains SEGA002 to SEGA004, featuring different C1 elements with 3′*tetA*, MG1655 TetA$^{OPT}$ was truncated using PCR products generated with oligos #107–#110 and #129–#131, followed by NiCl2 counter-selection. Promoters used in the SEGA landing pads were amplified from pET28a-AraH-GFP (PT7)[70], pPROTi[52] (PrhaBAD), pET-Duet1-Ptrc-GFPOpt (Ptrc)[68]. All C2–middle gadget combinations were transferred from strains with J23100 as C1 element to the other promoters. Oligos #111–#114, and #132 were used to amplify landing pad fragments containing the appropriate combinations of C2-gadget-*gfp*-L3S3P22-5′*tetA* (see Supplementary Figs. 8a–10a for visualization).

**Library design and construction**. TIR libraries of the different gadgets were constructed using a degenerated oligonucleotide specific for each gadget. In each degenerated oligonucleotide, the six nucleotides upstream of the start codon were changed to all possible codon combinations and the six nucleotides downstream of the start codon were changed to all possible synonymous codon combinations. All TIR libraries were constructed by PCR using Phusion Hot Start II Polymerase (Thermo Fisher Scientific, Waltham, MA, USA) or NeqXX7 DNA polymerase (self-made, not published). PCR products were purified and directly integrated into the genome of *E. coli* MG1655 J23100-3′*tetA* (SEGA001) as described in section "Genetic manipulation of *E. coli*" (see Supplementary Fig. 6a for visualization).

**Genetic manipulation of *E. coli***. All genetic manipulations of *E. coli* were performed using λ-Red recombineering. Strains harboring pSIM19[67] were prepared in a 250 mL shake flask containing 50 mL LB medium, supplemented with spectinomycin. The

cultures were inoculated 1:100 from an overnight culture and grown at 30 °C at 250 rpm until $OD_{600}$ of 0.4–0.5 was reached. Expression of the λ-Red proteins encoded on pSIM19 was induced by transferring the culture to 42 °C for 20 m in a shaking water bath. After induction, the culture was placed on ice for 15 m. Subsequently, the culture was transferred to a chilled 50 mL falcon tube and centrifuged at $4000 \times g$ for 5 m at 4 °C. For transformation of dsDNA or ssDNA for recombineering, cells were made electrocompetent by washing three times in ice-cold sterile water. 200 ng dsDNA or 200 μM single-stranded oligonucleotide were used for transformation by electroporation. dsDNA was obtained by PCR using Phusion Hot Start II Polymerase (Thermo Fisher Scientific, Waltham, MA, USA) or NeqXX7 DNA polymerase (self-made, not published) with oligonucleotides that contain homology region overhangs on both ends with a minimum size of 45 nucleotides, ideally 50 nucleotides or more. ssDNA oligos were designed using the online MAGE Oligonucleotide Design Tool (MODEST)[72] for targeting the lagging strand. *mCherry* and *rfp* were amplified from pmskl2 and pZE21-RFP, respectively (provided by Morten Sommer Lab). The nanobody was amplified from pET28a-nanobody-hp6-AmpR[70].

Cells were recovered with SOC medium at 30 °C at 800 rpm for 1 h, if not stated otherwise. In case *tetA* was integrated, cells were pelleted and plated onto LB agar or transferred to liquid LB medium supplemented with 25 μg/mL tetracycline for the selection of transformants. For $NiCl_2$ counter-selection, cells were recovered overnight at 30 °C in LB supplemented with spectinomycin. After recovery, cells were washed twice with sterile water and a dilution series was plated on M9-glucose minimal medium agar plates supplemented with 50 μM $NiCl_2$. In case of selection against *galK*, cells were recovered in 50 mL LB medium in a 250 mL shaking flask overnight at 30 degrees. Alternatively, cells can be recovered in 10 mL LB medium for 4.5 h, as described elsewhere[38]. After recovery, 1 mL of the cells were washed twice with 1X M9 salts and a dilution series was plated onto M63-glycerol minimal medium agar plates supplemented with 0.2% DOG. In case simultaneous induction of expression was necessary, 5 mM L-rhamnose or 1 mM IPTG were supplemented for the induction of PrhaBAD or PT7 and Ptrc, respectively.

Detailed protocols for genetic manipulation of SEGA landing pads utilizing the dual selection markers are provided at protocols.io (dx.doi.org/10.17504/protocols.io.bvxhn7j6).

**Fluorescence-activated cell sorting (FACS)**. For fluorescence-activated cell sorting, cultures were inoculated 1:100 from a stationary preculture and grown in LB medium supplemented with 10 μg/mL tetracycline for 7 h at 37 °C. For analysis, expression cultures were diluted (1:100) in 1X PBS. Flow cytometry measurements were performed on a SH800S Cell Sorter (Sony Biotechnology, San Jose, CA, USA) with excitation at 488 nm from a blue solid-state laser using the Sony Cell Sorter Software (v2.1.5). 10,000 cells were sorted into populations of low, medium, and high fluorescence intensity, each. The fraction with low fluorescence included cells up to the fifth percentile, the high fluorescent fraction those above the 95th percentile. To select for cells with medium fluorescence, a gate with arbitrary size was set at the mean fluorescence intensity value of the 5th and the 95th percentile. Sorted cells were grown in 2 mL LB supplemented with 10 μg/mL tetracycline overnight and 100 μL of a $10^6$ dilution was plated on LB agar. FlowJo (Treestar, Ashland, OR, USA) was used for data analysis. Flow cytometry dot plots used to identify the *E. coli* population of SEGA strains based on forward scattered (FSC) and side scattered (SSC) light are shown in Supplementary Figs. 13–17.

**Growth and fluorescence analysis**. Growth in liquid culture was assessed by growing the strains for 7 h in 24-deepwell plates. Growth was subsequently analyzed in 96-μl plates using the plate reader ELx808 (BioTek, Winooski, VT, USA) for $OD_{630}$ measurements only or Synergy H1 (BioTek, Winooski, VT, USA) for combined $OD_{630}$ and fluorescent measurements using the BioTek Gen5 software (v3.08). O/N cultures were diluted 1:100 in LB and incubated at 37 °C with continuous shaking. Cultures were induced with 1 mM IPTG or 5 mM L-rhamnose after 2 h, if necessary. In case of YidC-GFP production, cultures were induced with 1 mM IPTG after 2.5 h. Breathe-Easy® film (Sigma-Aldrich, St. Louis, MO, USA) was applied to minimize evaporation during the measurements. All strains were grown in triplicates, if not stated otherwise. For continuous growth measurements, growth was monitored every 30 m for at least 20 h. GFP fluorescence was measured from the bottom with excitation set to 485 nm and emission set to 528 nm. RFP fluorescence was measured from the bottom with excitation set to 585 nm and emission set to 615 nm. Growth and fluorescence curves were analyzed using Microsoft Excel and graphs shown were generated using GraphPad Prism version 9.1.0 (GraphPad Software, Inc., San Diego, CA, USA). GFP fluorescence values were normalized using a standard curve generated with sodium fluorescein dilutions ranging from 10 nM to 1 μM (see Supplementary Fig. 18).

**Flow cytometry**. For flow cytometry measurements, cultures were inoculated 1:100 from a stationary preculture and grown in LB medium supplemented with 25 μg/mL tetracycline for 7 h at 37 °C. At $OD_{600}$ 0.3–0.5 strains containing the T7 or *trc* promoter were induced with 1 mM IPTG and strains containing PrhaBAD were induced with 5 mM L-rhamnose. Flow cytometry measurements were performed on a MACSQuant VYB (Miltenyi Biotec, Bergisch Gladbach, Germany) with a blue 488 nm laser using the software MACSQuantify (v2.13.11). 10,000 cells were analyzed for each sample. FlowJo (Treestar, Ashland, OR, USA) was used for data analysis.

**Reporting summary**. Further information on research design is available in the Nature Research Reporting Summary linked to this article.

## Data availability

The FACS data generated in this study for Fig. 3 and Supplementary Figs. 6–10 have been deposited at FlowRepository[73] under accession codes FR-FCM-Z43R and FR-FCM-Z445. DNA sequences are provided with the source data of this paper. Relevant strains of the SEGA collection are deposited at the Belgian Co-ordinated Collections of Microorganisms (https://bccm.belspo.be) and the repository IDs can be found on www.sega-genomes.com/straincollection. The data for bar or dot graphs generated in this study are provided in the Source data file. Source data are provided with this paper.

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

## Acknowledgements

The authors thank Christopher Voigt (Massachusetts Institute of Technology, Cambridge, MA, USA) for providing the Marionette sensor collection upon request as well as Lars Ellgaard (Copenhagen University, Copenhagen, Denmark) and the Morten Sommer lab (Novo Nordisk Foundation Center for Biosustainability, Technical University of Denmark, Kgs. Lyngby, Denmark) for providing the plasmids pET39-Ub (His10) and pmskl2 and pZE21-RFP, respectively. Furthermore, we want to thank Cristina Hernández Rollán (Novo Nordisk Foundation Center for Biosustainability, Technical University of Denmark, Kgs. Lyngby, Denmark) for providing the plasmid pET28a-*sfgfp*. The authors also want to thank Joen Haahr Jensen, Jacob Søholm Mejlsted, and David Lokjær Faurdal for their support on constructing strains that contain promoters from the Marionette collection. The authors acknowledge funding by the Novo Nordisk Foundation (NNF20CC0035580) and by the Bioroboost project under EU Horizon 2020 research and innovation program under grant agreement N820699. A.K.E. was supported by grant no. NNF18CC0033664 as a fellow of the Copenhagen Bioscience PhD Programme.

## Author contributions

C.N.B., M.R., and A.K.E. contributed equally to this study. M.H.H.N. supervised this work.

## Competing interests

The authors declare no competing interests.
