## [Peer Review File · Nature Communications]

Reviewers' Comments:

Reviewer #1:

Remarks to the Author:

This paper presents SEGA, a standardized architecture for integrating DNA circuits into genomes. It complements SEVA, which is a standardized architecture for plasmid vectors. The authors correctly point out the importance of having a simple method for genome integrated genetic circuit design due to the various design challenges working with plasmid based circuits, such as variations in copy number and loss of the plasmid entirely over time. Genome integration provides a much more stable and predictable means of genetic circuit integration.

The proposed architecture is novel, and it will be useful to the synthetic biology community. The experimental results presented appear sound, and they support the conclusion that this architecture performs as desired. The authors are commended for the contribution to synthetic biology standards.

Where the article currently falls short is their use of existing standards. The authors do use the SBOL Visual standard for their diagrams, but they provide their DNA sequences in PDF form. This is roughly equivalent to not providing their sequences at all, since they are not in a usable format. The original SEVA paper provided their designs in the SBOL data format and shared them publicly via data repository (<https://sevahub.es>). The authors should do the same or this work is not going to be usable by the community. The authors have made their protocols available on protocols.io, and they have more information on a website (www.sega-genomes.com). The latter appears to only have a short video and nothing else so far. The authors state that their strains will be available at Addgene, which should be a requirement for publication.

In summary, this is an important development for the standardization of synthetic biology design. It will be highly useful to produce more reliable and predictable genetic designs. The authors though need to do more to make this work available to the user community before this work is ready for publication.

Chris Myers

Reviewer #2:

Remarks to the Author:

Key results

This paper introduces a method called Standardized Genome Architecture (SEGA) which uses a recombinering/lambda-red approach to integrate standardized DNA parts into a landing pad. Landing pads can be at different loci of the E. coli genome.

- The authors have demonstrated that SEGA enables genome engineering with cargo bricks and control bricks ranging in size from the J23100 promoter to the crtEBIY carotenoid biosynthesis pathway gene cluster.
- Integration of control bricks with different combinations of control elements (promoters, degenerate Shine-Dalgarno sequences, and degenerate 5' sequences of genes) and "gadgets" yield different phenotypes (in this case high, medium, and low GFP fluorescence).
- This approach of constructing controlled cargo bricks with different combinations of control elements was used to optimise expression of a "difficult-to-express" protein fused to GFP.
- Keio and MG1655 strains were used to demonstrate that landing pad integration efficiencies vary depending on loci and the background strain. This was also true for integration of a SEGA cargo brick carrying an rfp gene into landing pads at different loci.
- Finally, the authors have identified some correlation between observed GFP fluorescence and the distance of the gfp-harboring landing pad from the chromosomal origin of replication.

Validity

The data interpretation and conclusions made in this work are both valid and robust. The data is technically sound, obtained with appropriate techniques, analysed and interpreted carefully, and

presented in sufficient detail. In support of the conclusions, sufficiently strong evidence is provided for the authors' claims and all appropriate controls have been included.

Significance

The proposed landing pad architecture, standardized integration strategy, and strains generated within this work will be of value to the Molecular Biology community, but its significance is compromised by the existence of recombineering/lambda red (Sharan et al., 2009/ Thomason et al., 2007) and CRISPR/Cas9-mediated E. coli genome editing methods (Zhao et al., 2016). Both of these methods have been used for targeted integration of DNA fragments into the genome of E. coli. The main disadvantage of these methods compared to SEGA is that the homologous ends of the DNA fragments have to be designed to match the target locus. However, this means that a strain with an integrated DNA fragment can be engineered iteratively with additional fragments at additional loci with recombineering/lambda-red and CRISPR/Cas9-mediated strategies. The standardized regions of DNA homology used for SEGA mean that integration can only occur at one locus within a strain, and integration at multiple loci is not feasible. While these results will be important to the field, it is not likely that they will advance understanding in a way that will move the field forward.

Data and methodology

The quality of data and the quality of presentation in this manuscript and in the accompanying Supplementary Information are high. It is worth noting that fluorescence is not a direct measurement of gene expression (transcription or translation). However, fluorescence of GFP is widely used as a quantitative reporter of gene expression within the field of Molecular Biology. The approaches taken in the experiments presented in this work are valid.

Analytical approach

With the exception of Supplementary Figure 11, no statistical analysis has been conducted. Where appropriate, data points represent the means of three biological replicates and the error bars show the standard deviation. In research around this subject matter, this is considered a valid analytical approach. In contrast, it is not clear why datapoints of the three individual replicates in Figures 3e-h are shown rather than error bars.

Suggested improvements

Introduction

- Please explain the SEGA acronym in either/both the abstract or/and introduction.
- 57 – Suggested rewording "While on one hand standardization could restrict flexibility and creativity..."
- 71 – it seems odd to present an argument against standardization here.
- 77 – this could be interpreted as arguments against SEGA.
- 81 – is this an argument against standardization and therefore against SEGA too?
- Scope - 81 – references 26-32 refer to reduced/rewritten genomes, are there any additional ones that would cover the scope described?
- Context - Please provide some examples of genome integration strategies in E. coli, specifically recombineering/Lambda-red (Sharan et al., 2009/ Thomason et al., 2007) and CRISPR/Cas9-mediated E. coli genome engineering (Zhao, et al., 2016), how they work, their advantages/disadvantages, and why a method like SEGA would be better.
- Context - It would also be beneficial to explain the benefits of landing pads and give some examples of successful applications.

Philosophy and design of SEGA

- 110 – it would help the reader a little at this point to explain that C1 and C2 are promoters, Shine-Dalgarno sequences, and 5' sequences of genes.
- Figure 1b – should the landing pad brick have a "2>" icon above the mG?

SEGA enables highly simplistic genome engineering with green-white screening

- 184 – by “no false positives”, does this mean there were no green colonies?
- 185 – what about other 30%? If they are not false positives, what are they?
- In Figure 2e some colonies appear to be pink. Out of interest, not necessarily for inclusion in the manuscript, could this indicate lycopene production (crtEBI)? If this was due to inactivation of the crtY gene because selection against metabolic burden, would it have any correlation to the locus and landing pad used?
- Supplementary Figure 3c – out of interest, did crtEBI-3'tetA have a pink phenotype?
- 207 – should this say “absence” of rhamnose?

Construction and validation of a SEGA strain collection with different control elements

- Supplementary Figure 5 - it would help the reader to remind them that BCD means “bicistronic design”.
- Are Figures 3c and d the same as Supplementary Figures 6f and 7e? If so, please note on the legend for Figure 3 that this is PelBss TIR and if it is representative of the cell sorting and flow cytometry of TIR libraries shown in the Supplementary Information.
- In Figures 3e and g, please include the promoter name as in Figure 3f.

Multi-level control of a difficult-to-express gene is enabled by SEGA

- 339 – to help the reader, please describe what is on the single DNA fragment.
- Consider rewording for clarity, “...single DNA fragment harbouring RS2 and the truncated tetA gadget as homology regions was required to construct six different production strains”.

Efficiency of integration of SEGA landing pads varies between different genomic locations

- 378 - KEIO or Keio?
- Figure 5d – why does the y-axis represent CFU/ml overnight recovery culture?
- Figure 5d – how many replicates were included?
- 434 – should this say “Figure 5h”?

Additional thoughts for the authors to consider but not required for the manuscript.

- 667 – would it be possible to include the fluorescein standard curve?
- Activated cryostocks – does freezing a strain affect the efficiency of integration or gene expression levels?
- Would it be possible to include growth curves to accompany Figure 3 as indicators of cell health/metabolic burden?
- Was protein abundance (rather than fluorescence) ever investigated e.g. by SDS-PAGE? Were transcription levels measured e.g. by RT-qPCR?
- Is SEGA constrained to lambda-red? Could CRISPR/Cas9 be used?
- Can SEGA integration be done at multiple loci in same strain either all at once or iteratively?

Clarity and context

This manuscript is written with clarity, and it is accessible to those in fields related to Molecular Biology. As noted in the comments above, the introduction would benefit from stronger context. Specifically, it is strongly recommended that alternative genome engineering strategies in *E. coli* such as recombineering/Lambda-red (Sharan et al., 2009/ Thomason et al., 2007) and CRISPR/Cas9-mediated (Zhao, et al., 2016) approaches are briefly described, and their advantages and disadvantages discussed. It is also recommended that some examples of successful landing pads in *E. coli* are included, along with some indication of why SEGA is better.

References

- Please see previous notes in scope and context for the introduction.
- Please double check the references in the manuscript.
- 747 – no journal given.
- 779 - (2007).

- Thomason L, Court DL, Bubunencko M, Costantino N, Wilson H, Datta S, Oppenheim A.

Recombineering: genetic engineering in bacteria using homologous recombination. *Curr Protoc Mol Biol*. 2007 Apr;Chapter 1:Unit 1.16. doi: 10.1002/0471142727.mb0116s78. PMID: 18265390.

- Sharan SK, Thomason LC, Kuznetsov SG, Court DL. Recombineering: a homologous recombination-based method of genetic engineering. *Nat Protoc*. 2009;4(2):206-23. doi: 10.1038/nprot.2008.227. PMID: 19180090; PMCID: PMC2790811.
- Zhao D, Yuan S, Xiong B, Sun H, Ye L, Li J, Zhang X, Bi C. Development of a fast and easy method for *Escherichia coli* genome editing with CRISPR/Cas9. *Microb Cell Fact*. 2016 Dec 1;15(1):205. doi: 10.1186/s12934-016-0605-5. PMID: 27908280; PMCID: PMC5134288.

POINT-TO-POINT RESPONSE TO REVIEWER COMMENTS

We would like to thank the two reviewers for their positive and constructive feedback. Our responses to specific comments are highlighted below in blue font and with an (R:). Changes to the first version of the manuscript are highlighted in yellow in the accompanying marked-up file.

Reviewer #1 (Remarks to the Author):

This paper presents SEGA, a standardized architecture for integrating DNA circuits into genomes. It complements SEVA, which is a standardized architecture for plasmid vectors. The authors correctly point out the importance of having a simple method for genome integrated genetic circuit design due to the various design challenges working with plasmid based circuits, such as variations in copy number and loss of the plasmid entirely over time. Genome integration provides a much more stable and predictable means of genetic circuit integration.

The proposed architecture is novel, and it will be useful to the synthetic biology community. The experimental results presented appear sound, and they support the conclusion that this architecture performs as desired. The authors are commended for the contribution to synthetic biology standards.

Where the article currently falls short is their use of existing standards. The authors do use the SBOL Visual standard for their diagrams, but they provide their DNA sequences in PDF form. This is roughly equivalent to not providing their sequences at all, since they are not in a usable format. The original SEVA paper provided their designs in the SBOL data format and shared them publicly via data repository (<https://sevahub.es>). The authors should do the same or this work is not going to be usable by the community. The authors have made their protocols available on protocols.io, and they have more information on a website (www.sega-genomes.com). The latter appears to only have a short video and nothing else so far. The authors state that their strains will be available at Addgene, which should be a requirement for publication.

In summary, this is an important development for the standardization of synthetic biology design. It will be highly useful to produce more reliable and predictable genetic designs. The authors though need to do more to make this work available to the user community before this work is ready for publication.

Chris Myers

R: Thank you for the positive feedback. We already prepared the sequences in Genbank format for upload as a zipped file but were not able to upload this with the initial submission. Hopefully the editorial team can assist us in doing this. The SEGA webpage has been updated and the sequences can also be downloaded there. The strain collection has been submitted and sent to the microbial strain repository BCCM/GeneCorner and accessions numbers should be available soon and can be added

to the strain table prior to publication. This information has now been amended to the last paragraph in the discussion.

Reviewer #2 (Remarks to the Author):

Key results

This paper introduces a method called Standardized Genome Architecture (SEGA) which uses a recombineering/lambda-red approach to integrate standardized DNA parts into a landing pad. Landing pads can be at different loci of the *E. coli* genome.

- The authors have demonstrated that SEGA enables genome engineering with cargo bricks and control bricks ranging in size from the J23100 promoter to the crtEBIY carotenoid biosynthesis pathway gene cluster.
- Integration of control bricks with different combinations of control elements (promoters, degenerate Shine-Dalgarno sequences, and degenerate 5' sequences of genes) and "gadgets" yield different phenotypes (in this case high, medium, and low GFP fluorescence).
- This approach of constructing controlled cargo bricks with different combinations of control elements was used to optimise expression of a "difficult-to-express" protein fused to GFP.
- Keio and MG1655 strains were used to demonstrate that landing pad integration efficiencies vary depending on loci and the background strain. This was also true for integration of a SEGA cargo brick carrying an *rfp* gene into landing pads at different loci.
- Finally, the authors have identified some correlation between observed GFP fluorescence and the distance of the *gfp*-harbouring landing pad from the chromosomal origin of replication.

Validity

The data interpretation and conclusions made in this work are both valid and robust. The data is technically sound, obtained with appropriate techniques, analysed and interpreted carefully, and presented in sufficient detail. In support of the conclusions, sufficiently strong evidence is provided for the authors' claims and all appropriate controls have been included.

Significance

The proposed landing pad architecture, standardized integration strategy, and strains generated within this work will be of value to the Molecular Biology community, but its significance is compromised by the existence of recombineering/lambda red (Sharan et al., 2009/ Thomason et al., 2007) and CRISPR/Cas9-mediated *E. coli* genome editing methods (Zhao et al., 2016). Both of these methods have been used for targeted integration of DNA fragments into the genome of *E. coli*. The main disadvantage of these methods compared to SEGA is that the homologous ends of the DNA fragments have to be designed to match the target locus. However, this means that a strain with an integrated DNA fragment can be engineered iteratively with additional fragments at additional loci with recombineering/lambda-red and CRISPR/Cas9-mediated strategies.

The standardized regions of DNA homology used for SEGA mean that integration can only occur at one locus within a strain, and integration at multiple loci is not feasible. While these results will be important to the field, it is not likely that they will advance understanding in a way that will move the field forward.

R: Thank you for these comments. We agree that the novelty is based on the standardized landing pads, the simplistic protocols, and the strain collection - and not the recombineering system in itself. We also agree that a useful extension to SEGA would be additional landing pads in other loci. The current SEGA platform only supports integration of multiple fragments using either the novel split-*tetA* cycling between selection and counter-selection or by prior assembly of more complex constructs on a plasmid (again with the aid of split *tetA*).

Data and methodology

The quality of data and the quality of presentation in this manuscript and in the accompanying Supplementary Information are high. It is worth noting that fluorescence is not a direct measurement of gene expression (transcription or translation). However, fluorescence of GFP is widely used as a quantitative reporter of gene expression within the field of Molecular Biology. The approaches taken in the experiments presented in this work are valid.

Analytical approach

With the exception of Supplementary Figure 11, no statistical analysis has been conducted. Where appropriate, data points represent the means of three biological replicates and the error bars show the standard deviation. In research around this subject matter, this is considered a valid analytical approach. In contrast, it is not clear why datapoints of the three individual replicates in Figures 3e-h are shown rather than error bars.

R: For Figures 3e-h, the data is represented this way because it is the journal's requirement for this (bar chart type) data. This is probably also the case for X-Y plots, but the illustrations become very complex when plotting all individual data points in X-Y plots, so we chose instead to plot the average values with standard deviations indicated. To comply fully with the journal policy, we now also included the raw data for these plots as supplementary data in excel format (Supplementary File 2).

Suggested improvements

Introduction

- Please explain the SEGA acronym in either/both the abstract or/and introduction.

R: Thank you for pointing this out, we have included the explanation of the acronym in the abstract and the introduction.

- 57 – Suggested rewording “While on one hand standardization could restrict flexibility and creativity...”

R: Thank you for the suggestion, we changed the sentence.

- 71 – it seems odd to present an argument against standardization here.

R: We don't understand this comment. We write: “...standards are still in the innovator or early adopters' phase – and the field requires new impulses and simpler technologies that will convince an increasing number of researchers to adopt them”. This is an argument for developing standards.

- 81 – is this an argument against standardization and therefore against SEGA too?

R: We agree that this sentence is unclear. We replaced it with a paragraph providing more context to SEGA as also requested later by this reviewer.

- 77 – this could be interpreted as arguments against SEGA.

R: The sentence mentioned is “Plasmid-based approaches still dominate synthetic biology in bacteria, likely due to complicated genome engineering protocols...” and is provided as a motivation to simplify the current procedures as we then do with SEGA.

- Scope - 81 – references 26-32 refer to reduced/rewritten genomes, are there any additional ones that would cover the scope described?

R: It is correctly observed that we are missing references representing single-gene engineering examples (there will of course be many!). However, based on the comment above we chose to remove the sentence entirely so the references are no longer relevant.

- Context - Please provide some examples of genome integration strategies in E. coli, specifically recombineering/Lambda-red (Sharan et al., 2009/ Thomason et al., 2007) and CRISPR/Cas9-mediated E. coli genome engineering (Zhao, et al., 2016), how they work, their advantages/disadvantages, and why a method like SEGA would be better.

- Context - It would also be beneficial to explain the benefits of landing pads and give some examples of successful applications.

R: Thank you for pointing this out – we agree that the introduction could benefit from more detail. We have added a more detailed description of genome integration strategies in the introduction. We have not elaborated on the mechanistic advantages/disadvantages of those methods as we think that the same

advantages/disadvantages apply for SEGA. We do not see SEGA as a new method of genome engineering (as it requires the λ -Red system) but rather a platform to standardize genome engineering. However, to provide more context in the introduction, we have also provided specific examples of prior, more complex genome engineering approaches and on the use of landing pads.

Philosophy and design of SEGA

- 110 – it would help the reader a little at this point to explain that C1 and C2 are promoters, Shine-Dalgarno sequences, and 5' sequences of genes.

R: Thanks for pointing this out. We have rephrased the sentence and added a short explanation of the TIR together with a reference.

- Figure 1b – should the landing pad brick have a “2>” icon above the mG?

R: No, the landing pad brick only requires homology regions 1 and 3 for recombination, therefore homology region 2 is not indicated.

SEGA enables highly simplistic genome engineering with green-white screening

- 184 – by “no false positives”, does this mean there were no green colonies?

R: Yes, thanks for pointing this out. We clarified that false-positives are green fluorescent colonies.

- 185 – what about other 30%? If they are not false positives, what are they?

R: Colony PCRs confirmed that these clones carry the crtEBIY pathway. Sequencing of several such clones showed that sequence errors rendered the pathway non-functional. We added the information to the results section.

- In Figure 2e some colonies appear to be pink. Out of interest, not necessarily for inclusion in the manuscript, could this indicate lycopene production (crtEBI)? If this was due to inactivation of the crtY gene because selection against metabolic burden, would it have any correlation to the locus and landing pad used?

R: We interpret the colonies on the plate presented in Figure 2e as either white or orange and confirmed for some of the white colonies that the pathway was rendered non-functional by SNVs in different genes in the operon. We have at other times observed pink phenotypes, also while attempting to integrate the full crtEBIY pathway in one piece (see reply to the next comment). However, we did not systematically correlate a pink phenotype to mutations in crtY, different loci, or landing pad use. In these experiments, we used a high expressing landing pad (BCD with a high expressing TIR). It is certainly possible that using a landing pad e.g. with a low expressing TIR could reduce metabolic burden and increase the success rate.

- Supplementary Figure 3c – out of interest, did crtEBI-3'tetA have a pink phenotype?

R: We did observe clones with pink phenotypes during the crtEBIY pathway assembly process, both while performing iterative genome engineering as well as when integrating the whole pathway in one piece. In the latter case, our assumption is that the crtY gene was mutated in the strains showing a pink phenotype, although we never systematically sequenced these samples.

- 207 – should this say “absence” of rhamnose?

R: The phrasing in the manuscript is correct.

Construction and validation of a SEGA strain collection with different control elements

- Supplementary Figure 5 - it would help the reader to remind them that BCD means “bicistronic design”.

R: Thank you for this suggestion. We added the information in the figure legend of Supplementary Figure 5.

- Are Figures 3c and d the same as Supplementary Figures 6f and 7e? If so, please note on the legend for Figure 3 that this is PelBss TIR and if it is representative of the cell sorting and flow cytometry of TIR libraries shown in the Supplementary Information.

R: This is correct. The information was included in the figure legend.

- In Figures 3e and g, please include the promoter name as in Figure 3f.

R: The promoter names were added in Figure 3e and 3g

Multi-level control of a difficult-to-express gene is enabled by SEGA

- 339 – to help the reader, please describe what is on the single DNA fragment.
- Consider rewording for clarity, “...single DNA fragment harbouring RS2 and the truncated tetA gadget as homology regions was required to construct six different production strains”.

R: Thanks for pointing this out. We rephrased the sentence and hope that it is now better understandable for the reader.

Efficiency of integration of SEGA landing pads varies between different genomic locations

- 378 - KEIO or Keio?

R: We replaced “KEIO” with “Keio”.

- Figure 5d – why does the y-axis represent CFU/ml overnight recovery culture?

R: When a cargo is integrated within a SEGA landing pad, the *tetA* gene will be removed. This will allow for selection/growth on NiCl₂. However, cells will likely only be able to grow on NiCl₂ once all TetA transporters in the cell membrane are degraded or diluted away by growth for several generations. For this reason, we recover cultures, where cargo bricks have been integrated, over night in 5 mL medium before plating on NiCl₂.

- Figure 5d – how many replicates were included?

R: Three biological replicates were used. This is indicated in the end of the figure legend.

- 434 – should this say “Figure 5h”?

R: Yes, thank you for pointing out this mistake.

Additional thoughts for the authors to consider but not required for the manuscript.

- 667 – would it be possible to include the fluorescein standard curve?

R: Yes, this is a good suggestion. An example of a standard curve was included as Supplementary Figure 18.

- Activated cryostocks – does freezing a strain affect the efficiency of integration or gene expression levels?

R: We did not test this systematically but can state that we did not see a notable decrease in transformation and integration efficiency in activated cryostocks that had been frozen for several months, up to years.

- Would it be possible to include growth curves to accompany Figure 3 as indicators of cell health/metabolic burden?

R: We included OD₆₃₀ values measured after growing the cells for 7 hours as Supplementary Figure 11.

- Was protein abundance (rather than fluorescence) ever investigated e.g. by SDS-PAGE? Were transcription levels measured e.g. by RT-qPCR?

R: We did not check protein abundance on SDS-PAGE or transcription levels by RT-qPCR.

- Is SEGA constrained to lambda-red? Could CRISPR/Cas9 be used?

R: SEGA is not constrained to any specific recombination system. CRISPR/Cas9 can be used, either instead of e.g. the *tetA* gadget, or in combinations with other resistance markers as downstream gadgets. We initially used CRISPR/Cas9 in combination with Lambda-Red recombination to assemble the SEGA landing pad. Due to very low efficiencies, we would not recommend in any case to use CRISPR/Cas9 alone but combine it with the Lambda-Red system (or the Rac prophage RecET system).

- Can SEGA integration be done at multiple loci in same strain either all at once or iteratively?

R: Very interesting suggestion, which relates to one of the first points raised by this reviewer. As discussed above, a useful extension to SEGA would be additional landing pads in other loci. Such a landing pad would ideally have no homology to the other SEGA landing pad to avoid instability due to homologous recombination.

Clarity and context

This manuscript is written with clarity, and it is accessible to those in fields related to Molecular Biology. As noted in the comments above, the introduction would benefit from stronger context. Specifically, it is strongly recommended that alternative genome engineering strategies in *E. coli* such as recombineering/Lambda-red (Sharan et al., 2009/ Thomason et al., 2007) and CRISPR/Cas9-mediated (Zhao, et al., 2016) approaches are briefly described, and their advantages and disadvantages discussed. It is also recommended that some examples of successful landing pads in *E. coli* are included, along with some indication of why SEGA is better.

R: Thank you for pointing this out. As indicated above we added more information on alternative strategies and landing pads to the introduction. Furthermore, the first paragraph in the results section has been modified slightly and an additional sentence has been added to clarify how SEGA is different from the state-of-the-art.

References

- Please see previous notes in scope and context for the introduction.
- Please double check the references in the manuscript.
- 747 – no journal given.
- 779 - (2007).

- Thomason L, Court DL, Bubunenko M, Costantino N, Wilson H, Datta S, Oppenheim A. Recombineering: genetic engineering in bacteria using homologous recombination. *Curr Protoc Mol Biol.* 2007 Apr;Chapter 1:Unit 1.16. doi: 10.1002/0471142727.mb0116s78. PMID: 18265390.
- Sharan SK, Thomason LC, Kuznetsov SG, Court DL. Recombineering: a homologous recombination-based method of genetic engineering. *Nat Protoc.* 2009;4(2):206-23. doi: 10.1038/nprot.2008.227. PMID: 19180090; PMCID: PMC2790811.
- Zhao D, Yuan S, Xiong B, Sun H, Ye L, Li J, Zhang X, Bi C. Development of a fast and easy method for *Escherichia coli* genome editing with CRISPR/Cas9. *Microb Cell Fact.* 2016

Dec 1;15(1):205. doi: 10.1186/s12934-016-0605-5. PMID: 27908280; PMCID: PMC5134288.

R: We updated and corrected the reference list

Reviewers' Comments:

Reviewer #1:

Remarks to the Author:

The authors have now provided GenBank files on their website for their strains. They are also depositing their strains in a strain repository. Both of these steps are big improvements in my concerns raised about data sharing. I would like to further encourage the authors to convert their designs to SBOL and share them on an instance of SynBioHub. One natural location would be on seyahub.es (<http://seyahub.es>). Alternatively, they could deposit them on <https://synbiohub.org>. They should be able to create a Zip file and submit all the GenBank files at once. I would be happy to help with this activity, if they have any problems.

One issue they may face is that the GenBank files are exported from SnapGene. Unfortunately, SnapGene give each the LOCUS name Exported, which is not very illuminating. I understand it is a little tedious, but I suggest changing "Exported" in each GenBank file to the SEGA number to get a more meaningful name produced during the conversion to SBOL that SynBioHub performs. If they choose to upload to SevaHub, the authors might contact Angel Goni-Moreno who created this repository for SEVA to get his assistance as well.

Chris Myers

Reviewer #2:

Remarks to the Author:

The authors have addressed the points made in my initial review and therefore I am happy to recommend publication without further changes.

POINT-TO-POINT RESPONSE TO REVIEWER COMMENTS

We would again like to thank the two reviewers for their positive and constructive feedback. Our response to a specific comment is highlighted below in blue font and with an (R:). Changes to the latest version of the manuscript are marked-up in the accompanying file.

Reviewer #1 (Remarks to the Author):

The authors have now provided GenBank files on their website for their strains. They are also depositing their strains in a strain repository. Both of these steps are big improvements in my concerns raised about data sharing. I would like to further encourage the authors to convert their designs to SBOL and share them on an instance of SynBioHub. One natural location would be on sevahub.es (<http://sevahub.es>). Alternatively, they could deposit them on <https://synbiohub.org>. They should be able to create a Zip file and submit all the GenBank files at once. I would be happy to help with this activity, if they have any problems.

One issue they may face is that the GenBank files are exported from SnapGene. Unfortunately, SnapGene give each the LOCUS name Exported, which is not very illuminating. I understand it is a little tedious, but I suggest changing "Exported" in each GenBank file to the SEGA number to get a more meaningful name produced during the conversion to SBOL that SynBioHub performs. If they choose to upload to SevaHub, the authors might contact Angel Goni-Moreno who created this repository for SEVA to get his assistance as well.

Chris Myers

(R:) We converted all the 104 sequences to SBOL and uploaded them to synbiohub.org. The collection needs to be approved by a curator to be public. We have reached out to Chris Myers and has agreed to make the collection public.

Reviewer #2 (Remarks to the Author):

The authors have addressed the points made in my initial review and therefore I am happy to recommend publication without further changes.